# Relationship between Fear-Avoidance Beliefs and Muscle Co-Contraction in People with Knee Osteoarthritis

**DOI:** 10.3390/s24165137

**Published:** 2024-08-08

**Authors:** Takanori Taniguchi, So Tanaka, Tomohiko Nishigami, Ryota Imai, Akira Mibu, Takaaki Yoshimoto

**Affiliations:** 1Department of Physical Therapy, Faculty of Medical Science, Fukuoka International University of Health and Welfare, Fukuoka 814-0001, Japan; taniguchi@s.takagigakuen.ac.jp; 2Department of Clinical Research Center, Fukuoka Orthopaedic Hospital, Fukuoka 815-0063, Japan; loose_no_6@ybb.ne.jp; 3Department of Physical Therapy, Faculty of Health and Welfare, Prefectural University of Hiroshima, Hiroshima 723-005, Japan; 4Department of Rehabilitation, Osaka Kawasaki Rehabilitation University, Osaka 597-0104, Japan; imair@kawasakigakuen.ac.jp; 5Department of Physical Therapy, Konan Women’s University, Hyogo 658-0001, Japan; a_mibu@konan-wu.ac.jp; 6Department of Orthopaedic, Fukuoka Orthopaedic Hospital, Fukuoka 815-0063, Japan; tyoshimoto.fukuokask@gmail.com

**Keywords:** co-contraction, fear-avoidance beliefs, kinesiophobia, pain catastrophizing, knee osteoarthritis

## Abstract

Excessive muscle co-contraction is one of the factors related to the progression of knee osteoarthritis (OA). A previous study demonstrated that pain, joint instability, lateral thrust, weight, and lower extremity alignment were listed as factors affecting excessive co-contraction in knee OA. However, this study aimed to assess the association between fear-avoidance beliefs and muscle co-contraction during gait and stair climbing in people with knee OA. Twenty-four participants with knee OA participated in this cross-sectional study. Co-contraction ratios (CCRs) were used to calculate muscle co-contraction during walking and stair climbing, using surface electromyography. Fear-avoidance beliefs were assessed by the Tampa Scale for Kinesiophobia-11 (TSK-11) for kinesiophobia and the Pain Catastrophizing Scale (PCS) for pain catastrophizing. Secondary parameters that may influence co-contraction, such as degree of pain, lateral thrust, weight, and lower extremity alignment, were measured. The relationships between the CCR during each movement, TSK-11, and PSC were evaluated using Spearman’s rank correlation coefficient and partial correlation analysis, adjusted by weight and lower extremity alignment. Partial correlation analysis showed a significant correlation only between medial muscles CCR and TSK-11 during stair descent (r = 0.54, *p* < 0.05). Our study revealed that kinesiophobia could be associated with co-contraction during stair descent in people with knee OA.

## 1. Introduction

Knee osteoarthritis (OA) is one of the most common chronic degenerative joint diseases in the elderly [1], with approximately 30% of people over the age of 45 years having radiological evidence of knee OA [2]. Clinical symptoms of knee OA include arthralgia, joint deformity, limited range of motion, and muscle weakness [3]. These clinical symptoms could affect gait and the ability to climb stairs [4]. As the disease progresses, daily activities such as walking and climbing stairs become difficult [5]. Therefore, it is important for people with knee OA to prevent the progression of the condition and to maintain their ability to walk and climb stairs.

Excessive muscle co-contraction of quadriceps and hamstrings is known as one of the factors related to the progression of OA and disability [6,7]. Muscle co-contraction is defined as the muscle activity in which the agonist and antagonist muscles contract simultaneously [8]. Appropriate co-contraction serves to stabilize and protect the joint [9], but excessive co-contraction increases joint contact pressure and the risk of progressive articular cartilage degeneration [7]. In addition, in people with knee OA, prolonged co-contraction increases the likelihood of conversion to total knee arthroplasty [6]. Excessive co-contraction also causes difficulty in gait and stair climbing in people with knee OA [10]. Co-contraction is an important component of gait and stair climbing disability in people with knee OA, and thus, it should be appropriately corrected.

Although several previous studies demonstrated that pain [11,12], lateral thrust [9,13], weight gain [14], and lower extremity alignment [15] are listed as factors affecting excessive co-contraction of knee OA, no definitive conclusion has been drawn. In addition to these factors, fear-avoidance beliefs may be considered as a factor influencing excessive co-contraction. Fear-avoidance beliefs can be explained as avoidance of behaviors that may cause pain and pessimistic or catastrophic interpretations of pain [16], including kinesiophobia and pain catastrophizing. Kinesiophobia is an excessive, irrational, and debilitating fear of physical movement and activity that results from a sense of vulnerability to painful injury or re-injury [17]. Pain catastrophizing is a negative mental state that occurs when experiencing actual or anticipated pain [18]. People with knee OA often have fear-avoidance beliefs, such as kinesiophobia and pain catastrophizing [19], which have been reported to be associated with performance during walking and stair climbing [20,21,22,23]. In people with knee OA, kinesiophobia has been reported to be associated with speed during gait [20] and self-rated difficulty during stair climbing [21]. It has also been reported that pain catastrophizing decreases postural stability [22] and the ability to climb stairs [23]. Recent studies have shown an association between fear-avoidance beliefs and altered motor control in patients with low back pain, which can increase muscle co-contractions [24,25]. It is inferred that this is due to potential interactions between fear-avoidance belief, biomechanical mechanisms, and supraspinal processes [26]. Karayannis et al. reported that fear-avoidance belief alters mechanical properties of the spine, including trunk stiffness by increasing trunk muscle excitability (tight control) in chronic low back pain [24]. The tight motor control strategy, which is characterized by high levels of muscle tension, has been associated with cortical reorganization that may impede or delay the restoration of normal motor control patterns [27]. We hypothesized that there would be a relationship between fear-avoidance beliefs and co-contraction during gait and stair climbing in people with knee OA. However, it is unclear whether kinesiophobia and/or pain catastrophizing affect co-contraction during gait and/or stair climbing. If the relationship between fear-avoidance beliefs and co-contraction is clarified, there may be more options for assessment and treatment of co-contraction. The purpose of this study was to clarify the relationship between fear-avoidance beliefs and co-contraction during gait and stair climbing in people with knee OA.

## 2. Materials and Methods

### 2.1. Study Design and Participants

Participants in this cross-sectional study were 20 people with knee OA recruited at the orthopedic hospital, between December 2020 and September 2021. The method of recruiting participants was through a direct announcement by a physical therapist to outpatients with knee OA, who met the inclusion criteria. The inclusion criteria were (1) men and/or women aged 50 to 80 years, who were diagnosed with knee OA, according to the clinical or radiographic classification criteria provided by the American College of Rheumatology, and (2) those with proper gait and with the ability to climb stairs themselves. When OA was present in both knees, the more symptomatic knee was selected as the target limb. The exclusion criteria were (1) neuromuscular diseases such as Parkinson’s disease or stroke; (2) cardiovascular disorders; (3) rheumatoid arthritis; (4) knee and/or spine trauma or surgery in the past year; and (5) difficulty in climbing 8 cm high stairs for pain, one step at a time. We checked whether participants could perform ascending and descending movements on stairs with a height of 8 cm and excluded those who were unable to perform stair climbing.

### 2.2. Testing Protocol

Initially, an assessment of fear-avoidance beliefs was performed with the Tampa Scale for Kinesiophobia-11 (TSK-11) and the Pain Catastrophizing Scale (PCS), using a questionnaire. Later, surface electromyography (EMG) was performed to randomly measure walking, stair climbing, and stair descent. Walking was performed three times on a 10 m sidewalk, at an optimal speed without rest [28]. The participants were instructed to walk in a usual manner. Data from five stable walking cycles were used for analysis. Stair climbing was performed three times without rest at an optimal speed, in the form of one step per foot [29]. The participants were instructed to ascend (or descend) the stairs in the usual manner and from the affected side. The staircase used was the one installed at the measurement location, with a height of 8 cm, width of 90 cm, depth of 30 cm, and five steps. Other secondary parameters that may influence co-contraction, such as pain, lateral thrust, weight gain, lower extremity alignment, and knee extension strength, were also measured.

### 2.3. Data Collection and Processing

Muscle activity during gait and stair climbing was measured using a 4-channel wireless surface EMG (Myomuscle, Noraxon, Scottsdale, AZ, USA) with 16-bit resolution, a common-mode rejection ratio >100 dB, and input impedance >100 MΩ. The raw waveforms were sampled at 1500 Hz with a 500 Hz low-pass filter. Ag/AgCl disk electrodes (Blue Sensor M-00-S, Ambu, Ballerup, Denmark; electrode diameter: 34 mm; shape: circular; distance between electrodes: 30 mm) were applied according to the Surface ElectroMyoGraphy for the Non-Invasive Assessment of Muscle recommendations. Electrodes were applied to the vastus medialis, vastus lateralis, semitendinosus, and biceps femoris muscles (Figure 1) [30]. The site of electrode application was assessed by palpation to confirm muscle contraction; the skin was treated with spirit. To identify the stance phase of gait and stair climbing movements, inertial measurement unit (IMU) sensors (Myomotion, Noraxon, Scottsdale, AZ, USA) synchronized with wireless surface EMGs were attached to the dorsal surfaces of the right and left feet (Figure 1). To measure lateral thrust, IMU sensors were attached to the right and left proximal tibia and acceleration was sampled at 100 Hz.

MyoResearch (Noraxon, Scottsdale, AZ, USA) was used to analyze the myoelectric signals. Myoelectric signals were band-pass filtered at 20–500 Hz to remove noise artifacts [7,31,32], and the root mean square was calculated on a 100 ms time axis. Additionally, each myoelectric signal was normalized by the peak dynamic method (PDM) to adjust for individual differences. The PDM is a method of normalizing myoelectric signals by dividing them by the maximum myoelectric signal during the same movement and has been confirmed to be as accurate as maximum voluntary contraction (MVC) [33]. Therefore, normalization by the PDM is appropriate when it is difficult to perform MVC due to severe knee OA, as in the participants of this study, or when MVC is smaller than the muscle activity due to gait [33]. Also, crosstalk between the quadriceps and hamstrings was confirmed using the crosstalk index [34,35,36] and was approximately 5% during knee extension and approximately 15% during knee flexion. The stance phase of gait and stair climbing was defined from the peak value of the vertical component of the foot acceleration data [37], and the time was normalized with 100 frames for the stance phase to align each movement time. The processed myoelectric signals data were averaged over five gait cycles for gait and three trials for stair climbing.

### 2.4. Co-Contraction Data

Co-contraction ratios (CCRs) were obtained for the degree of co-contraction, on the basis of the method of Dixon et al. [9]. This method is often used to detect CCRs [6,29]. CCRs are a measure of the degree of simultaneous contraction of the agonist and antagonist muscles, and are obtained by finding myoelectric signals with low (lower EMGi) and high activities (higher EMGi) over each data frame [9] (1). Higher CCRs suggest a higher rate of simultaneous use of the agonist and antagonist muscles [10]. CCRs of the medial (vastus medialis and semitendinosus) and lateral muscles (vastus lateralis and biceps femoris) were evaluated. The formula for calculating the CCR is as follows:(1)$$CCR=\sum_{i=1}^{n}\left(lowerEMGi/higerEMGi\right)(lowerEMGi+higherEMGi)/n$$

### 2.5. Assessment Questionnaires

#### 2.5.1. Tampa Scale for Kinesiophobia-11

Kinesiophobia was measured using the Tampa Scale for Kinesiophobia-11 (TSK-11). The TSK-11 is a questionnaire that investigates the psychometric properties of the original TSK and carefully selects the items of particular importance. It is a short questionnaire consisting of 11 items, with the same level of accuracy as the original TSK [38]. Items in TSK-11 are scored from 1 (strongly disagree) to 4 (strongly agree) points. Thus, the total TSK-11 score ranges from 11 to 44 points, with higher scores indicating greater fear of pain, movement, and injury.

#### 2.5.2. Pain Catastrophizing Scale

Pain catastrophizing was measured using the Pain Catastrophizing Scale (PCS) [34,39]. The PCS is a valid and reliable self-reporting questionnaire that uses 13 items to describe the thoughts and feelings that a person may experience when in pain [18]. Items on the PCS are scored from 0 (never) to 4 (always) points. Thus, the total PCS score ranges from 0 to 52 points, with higher scores indicating a more intense level of pain [18].

### 2.6. Secondary Parameters That May Affect Co-Contraction

#### 2.6.1. Degree of Pain

The degree of pain was assessed immediately after each movement was performed, using the visual analog scale. The participant was instructed to record the subjective pain level, with no pain being 0 and the highest level of pain being 100.

#### 2.6.2. Lateral Thrust

Lateral thrust is dynamic knee instability in the forehead plane during the early stance phase of gait [40]. Lateral thrust was defined by acceleration data measured by the IMU sensors attached to the proximal tibia and expressed as the average of the peak lateral acceleration in the early stance phase of walking.

#### 2.6.3. Weight

Weight was evaluated by body mass index (BMI) to account for individual differences. BMI was defined as the body weight in kg divided by the square of the body height in meters.

#### 2.6.4. Alignment of the Lower Extremities

Lower extremity alignment was assessed using femorotibial angle (FTA). FTA is the angle between the femur and tibia and was measured using forehead radiographs of the lower extremity, taken in the standing position.

#### 2.6.5. Knee Extension Strength

Maximal voluntary isometric knee extension strength was measured with the participant sitting, using a calibrated dynamometer (Micro FET 2; Hoggan Scientific, LLC., Salt Lake City, UT, USA). The thigh was fixed to the seat at the distal femur. The moment arm was attached to the tibia just above the malleoli. The knee and hip angles were fixed at 90°. The subjects performed as many maximal actions until the peak value no longer increased. The results were divided by body weight and expressed as N·m/kg.

### 2.7. Statistical Analyses

The sample size was calculated by performing a power analysis, with an estimated effect size of 0.6, power of 0.8, and α of <0.05, referring to the results of the analysis at mid-course [41,42]. The power analysis results indicated that at least 17 participants were needed.

Each measurement was subjected to a Shapiro–Wilk test, to confirm the normality of the distribution. First, to confirm sex differences in co-contraction and fear-avoidance beliefs, we compared females and males using the Mann–Whitney U test. To examine the relationship between co-contraction and fear-avoidance beliefs during gait, stair ascent, and stair descent, we performed univariate correlation analysis between TSK-11 and CCR and between PCS and CCR in each movement, using Spearman’s rank correlation coefficient. We also performed correlation analysis between CCR and visual analog scale for pain, the peak value of lateral acceleration, BMI, FTA, and knee extension strength using Spearman’s rank correlation coefficient to examine the relationship between co-contraction and secondary parameters that may affect co-contraction. Further, to clarify the relationship between co-contraction and fear-avoidance beliefs, we analyzed the partial correlations between CCR and TSK-11 and between CCR and PCS during each movement, adjusted for BMI and FTA. The BMI and FTA were chosen as confounding variables because previous studies have reported that the BMI and FTA affect co-contraction [14,15], and a prior Spearman’s rank correlation analysis showed a high correlation. Statistical analysis was performed using SPSS statistics ver.26 (IBM SPSS Statistics, Armonk, NY, USA). Significance was set at *p* < 0.05.

## 3. Results

The characteristics of the participants in this study are shown in Table 1. No gender differences were found in co-contraction and fear-avoidance beliefs (Table 2). The correlation coefficients between TSK-11 score and CCR and between PCS score and CCR for gait, stair ascent, and stair descent are shown in Table 3. The TSK-11 score was positively correlated with CCR during stair descent (medial muscles, *r* = 0.50; lateral muscles, *r* = 0.48, *p* < 0.05) (Figure 2). Co-contraction during stair ascent was significantly correlated with FTA (medial muscles, r = 0.48, *p* < 0.05; lateral muscles, r = 0.57, *p* < 0.01), whereas co-contraction during stair descent was significantly correlated with BMI (lateral muscles, r = 0.52, *p* < 0.05) and FTA (medial muscles, r = 0.48, *p* < 0.05; lateral muscles, r = 0.55, *p* < 0.05) (Table 4). Partial correlation analysis adjusted for BMI and FTA, showed a significant correlation only between medial muscles, CCR, and TSK-11 during stair descent (r = 0.54, *p* < 0.05) (Table 5).

## 4. Discussion

An association was found between kinesiophobia and medial/lateral muscle co-contraction during stair descent. However, after adjusting the BMI and FTA, kinesiophobia was found to be associated with only medial muscle co-contraction. During hop test landings in anterior cruciate ligament (ACL)-injured patients, a high level of kinesiophobia had increased co-contraction compared with a low level of kinesiophobia [43]. It has been suggested that this pattern of muscle activity may protect the knee joint by stiffening the joint and by limiting the forward movement of the tibia [44]. It is possible that the co-contraction during stair descent in people with knee OA is due to the movement strategies for fear of impact during landing, as well as the hop test in ACL-injured patients. Another study showed that increased lower extremity muscle activity during a single leg squat with similar biomechanics to stair decent may be influenced by kinesiophobia via decreases in angular displacement of the knee in patients with knee OA [45]. This mechanism is due to pre-programmed actions to decrease joint instability, a hallmark of knee OA, and kinesiophobia may have altered this pre-programming, resulting in excessive muscle contraction [25,45].

Co-contraction during stair descent was associated with TSK-11 but was not associated with PCS. This finding is congruent with that of a previous study, which stated that only TSK was associated with trunk stiffness in patients with chronic lower back pain [24]. As suggested by the previous study, no relationship between kinematic change and PCS may reflect the context and interpretation of the questionnaire. The TSK included questions that directly reflect fear and avoidance of movement and motion, whereas the PCS included questions primarily about catastrophic/pensive thoughts and questions about fear, worry, and anxiety [18], which may have influenced the results.

There were no significant associations between co-contraction and both TSK-11 and PCS during gait and stair ascent. Stair descent requires an increased load on the knee joint and stronger centrifugal contraction of the quadriceps and hamstrings than gait or stair ascending, and it has different movement characteristics [46]. Stair descent is a more difficult movement than walking or stair ascending [42], with a greater load on the knee joint [47]. Previous studies showed that kinesiophobia did not affect the kinematics of walking [48] but did affect those of more difficult tasks like jumping in patients with ACL injury [43]. This result suggested that kinesiophobia may be influenced by the difficulty of the movement. Thus, it is possible that only co-contraction during stair descent was associated with kinesiophobia.

Both medial and lateral muscle co-contractions during stair descent were associated with kinesiophobia. However, after excluding the effect of the BMI and FTA as confounders, only medial muscle co-contraction was associated with kinesiophobia. These findings suggest that the correlation between lateral muscle co-contraction and kinesiophobia without adjustment may be spurious. Factors affecting the difference between medial and lateral muscle co-contractions on lateral thrust, lower extremity alignment, and the degree of cartilage damage have been examined, but no definitive conclusions have been drawn [7,13,15]. Our results imply that the BMI and FTA have more impact on lateral muscle co-contractions than on medial ones.

The lack of association between the degree of pain and co-contraction in people with knee OA may have been influenced by the measurement methods and settings. Boyer et al. [12] showed that people whose pain increased after 20 min of walking have greater co-contraction in the late stance phase of gait. Continuous walking can increase knee joint load and pain, which can alter the biomechanical strategy for gait [12,49]. However, we assessed pain intensity after a 10 m walk, so the walking time was only a few tens of seconds. It is possible that the difference in walking time affected the difference in results. Moreover, the lack of association between co-contraction and pain during stair ascent and descent may have been influenced by the stair height. Different stair heights produce different mechanical loads on the knee joint, resulting in varying degrees of pain [50]. The stair height used in this study was lower than the stair height used in a previous study [21]. Therefore, it is possible that co-contraction and pain during stair ascent and descent were not related in this study. The lack of association of lateral thrust with co-contraction in people with knee OA may have been influenced by the severity of OA. Lateral thrust in people with knee OA is related to OA severity [51,52]. The dynamics of the lateral thrust may have differed between the previous study and the present study [9], and this may have influenced the co-contraction.

Proper co-contraction in patients with knee OA helps to stabilize the joint [9]. Furthermore, adequate co-contraction of the lateral muscles can prevent the progression of OA [7]. However, excessive co-contraction in knee OA increases joint surface pressure and risks progressive degeneration of articular cartilage [7]; thus, it must be properly treated. Although dry needling and the Alexander technique have been reported as interventions for knee OA co-contraction [53,54], the effect is not significant. This is because psychosocial factors affecting co-contraction have not been revealed to date and have not been included as the target of the intervention. Interventions directly targeting kinesiophobia have shown that individuals with pain can successfully experience and habituate to movements and activities that they might normally avoid [55]. Since the present study revealed that co-contraction during stair descent is associated with kinesiophobia, it is expected that interventions for kinesiophobia may influence the improvement in co-contraction.

A major strength of this study is that we conducted a partial correlation analysis that adjusted for confounding factors added to the simple correlations analysis, thereby possibly reducing the risk of pseudo-correlation. The result of partial correlation analysis revealed that the correlation between lateral muscle co-contraction and kinesiophobia without adjustment may be spurious. Additionally, aside from the factors (pain, lateral thrust, weight gain, and lower extremity alignment) [11,13,14,15] identified in previous studies, we identified kinesiophobia as a possible new target for excessive muscle co-contraction during stair descent. A systematic review and randomized controlled trial showed that multidisciplinary rehabilitation [56], pain neuroscience education [57] with exercise, and graded exposure therapy [58] improve pain, disability, and kinesiophobia. Any of these interventions may lead to improved movement when stair climbing is difficult.

This study had several limitations. First, we did not limit the use of a cane, handrail, or the speed of movement to reflect the daily life of people with knee OA as much as possible. Therefore, more detailed results may be obtained if the measurement conditions are strictly set. Second, the study was cross-sectional; therefore, we were not able to conclude the causal relationship between kinesiophobia and muscle co-contraction. Third, we did not analyze kinetics and joint moments using ground floor reaction. Future studies using ground reaction force will provide more insight into the relationship between muscle co-contraction and kinesiophobia. Fourth, the height of the stairs (8 cm) in this study was lower than that in a previous study (17 cm). Because it may not be possible for participants to perform stair claiming at normal heights and stair climbing may cause pain, we chose an 8 cm staircase. Indeed, nine participants could not perform 8 cm stair climbing because of pain. Therefore, results may differ with a normal-height staircase in participants with mild OA. Fifth, this study did not include normal controls. It was ethically impossible to perform X-ray examinations to measure FTA in normal controls. FTA is an important evaluation in this study, because previous studies have reported that the FTA affects co-contraction. This study did not compare subjects with normal controls, so the differences in mechanisms between people with OA and healthy people are unclear. Sixth, the thickness of the subcutaneous fat in the thigh is a factor that affects the surface EMG signal [59]. However, in this study, the thickness of the subcutaneous fat on the thighs was not measured. Therefore, the thickness of the subcutaneous fat may have influenced the EMG data. Seventh, the fear of falling was not considered. Because there were already many evaluation items in this study, the burden on the subjects was prioritized, and the fear of falling was not evaluated. However, fear of falling may affect excessive co-contraction. Therefore, measurements of the Falls Efficacy Scale (FES) and Activity-Specific Balance Confidence (ABC) scale should have been conducted. Eighth, there is a possibility that crosstalk may be affecting the EMG data. Electrodes were applied according to SENIAM to minimize crosstalk. However, the crosstalk index between the quadriceps and hamstrings was approximately 5% during knee extension and approximately 15% during knee flexion. Therefore, crosstalk may have influenced the EMG data.

## 5. Conclusions

This study aimed to assess the association between fear-avoidance beliefs and muscle co-contraction during gait and stair climbing in people with knee OA. Our study revealed that kinesiophobia could be associated with co-contraction during stair descent in people with knee OA.

## Figures and Tables

**Figure 1 sensors-24-05137-f001:**
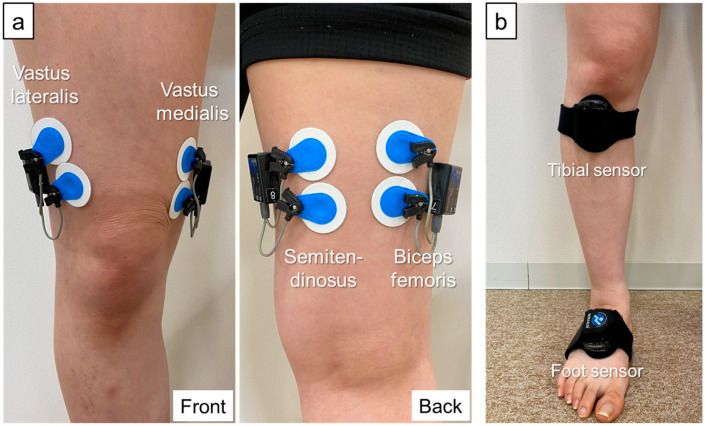
EMG electrode and IMU sensor placement. (**a**) EMG electrode placement. Vastus medialis—distal two-thirds position on the line connecting the anterior superior iliac spine and the joint space in front of the anterior border of the medial ligament. Vastus lateralis—distal 2/3 position on the line connecting the anterior superior iliac spine and the lateral patella. Semitendinosus—midway between the ischial tuberosity and medial epicondyle of the tibia. Biceps femoris—midway between the ischial tuberosity and lateral epicondyle of the tibia. (**b**) IMU sensor placement. The tibial sensors are banded to the front of the bilateral proximal lower legs. Lateral thrust is defined from the acceleration data of the lower leg sensor in the lateral direction. The foot sensors are banded to the dorsum of the foot on both sides. The stance phase is defined from the vertical acceleration data of the foot sensor. EMG, electromyography; IMU, inertial measurement unit.

**Figure 2 sensors-24-05137-f002:**
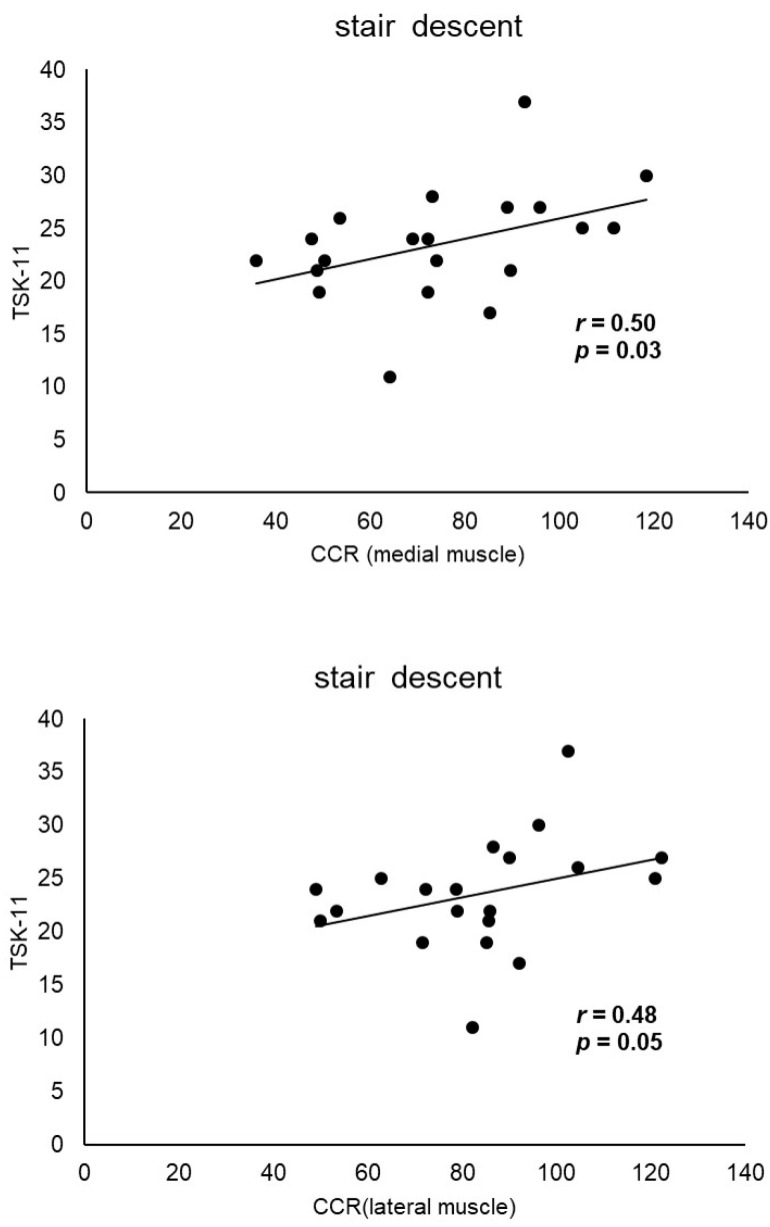
Correlation between the CCR and TSK-11 score during stair descent. The TSK-11 score is positively correlated with the CCR during stair descent (medial muscles, r = 0.50; lateral muscles, r = 0.48, *p* < 0.05). TSK-11, Tampa Scale for Kinesiophobia-11; CCR, co-contraction ratio. Circles: CCR and TSK-11 scores during stair descent were plotted; lines: regression line.

**Table 1 sensors-24-05137-t001:** Characteristics of the participants and the summary of measurements.

Variable			Mean ± SD
Age (year)			69.4 ± 7.4
Sex (female/male)			15/5
Height (cm)			155.2 ± 9.0
Weight (kg)			65.5 ± 12.2
Kellgren–Lawrence grade			II: 1, III: 8, IV: 11
CCR (%)	Gait	Medial muscles	62.5 ± 23.1
		Lateral muscles	75.6 ± 19.8
	Stair ascent	Medial muscles	60.6 ± 18.6
		Lateral muscles	81.5 ± 28.1
	Stair descent	Medial muscles	74.8 ± 23.2
		Lateral muscles	83.5 ± 20.5
TSK-11 score			23.6 ± 5.3
PCS score			20.6 ± 10.4
Pain VAS score (mm)			
Gait			23.0 ± 18.4
Stair ascent			27.1 ± 22.9
Stair descent			42.2 ± 29.9
Lateral acceleration (G)			0.8 ± 0.4
BMI			27.1 ± 4.0
FTA (°)			184.7 ± 3.7
Knee extension strength (N·m/kg)			27.4 ± 15.7

CCR: co-contraction ratios; medial muscles: vastus medialis/semitendinosus; lateral muscles: vastus lateralis/biceps femoris; TSK-11: Tampa Scale for Kinesiophobia-11; PCS: Pain Catastrophizing Scale; VAS: visual analog scale; BMI: body mass index; FTA: femorotibial angle; SD: standard deviation.

**Table 2 sensors-24-05137-t002:** Gender differences in co-contraction and fear-avoidance beliefs.

Variable			Female			Male			*p*-Value
TSK-11 score			24.0	±	5.9	22.2	±	3.4	0.38
PCS score			20.9	±	10.2	19.8	±	12.3	0.79
CCR (%)	Gait	Medial muscles	62.2	±	24.5	63.4	±	20.8	0.83
		Lateral muscles	76.8	±	19.8	71.7	±	21.3	0.69
	Stair ascent	Medial muscles	58.6	±	20.3	66.7	±	11.9	0.41
		Lateral muscles	84.1	±	24.1	73.8	±	40.2	0.90
	Stair descent	Medial muscles	77.7	±	24.8	65.9	±	16.6	0.32
		Lateral muscles	84.3	±	23.3	81.0	±	8.5	0.63

**Table 3 sensors-24-05137-t003:** Correlation of co-contraction with fear-avoidance beliefs during each movement.

Variable	CCR											
	Gait				Stair Ascent				Stair Descent			
	Medial Muscles		Lateral Muscles		Medial Muscles		Lateral Muscles		Medial Muscles		Lateral Muscles	
	Rho	*p*-Value	Rho	*p*-Value	Rho	*p*-Value	Rho	*p*-Value	Rho	*p*-Value	Rho	*p*-Value
TSK-11 score	0	0.99	−0.01	0.96	−0.07	0.76	0.25	0.27	0.50	0.03 *	0.48	0.04 *
PCS score	0.1	0.67	0.09	0.69	0.27	0.24	0.22	0.33	0.25	0.28	0.08	0.73

*: *p* < 0.05.

**Table 4 sensors-24-05137-t004:** Correlation of co-contraction with secondary parameters during each movement.

Variable	CCR											
	Gait				Stair Ascent				Stair Descent			
	Medial Muscles		Lateral Muscles		Medial Muscles		Lateral Muscles		Medial Muscles		Lateral Muscles	
	Rho	*p*-Value	Rho	*p*-Value	Rho	*p*-Value	Rho	*p*-Value	Rho	*p*-Value	Rho	*p*-Value
Pain VAS score	−0.08	0.75	0.23	0.33	0.02	0.94	−0.06	0.8	0.05	0.83	−0.18	0.46
Lateral acceleration	0.12	0.62	0.07	0.78	―	―	―	―	―	―	―	―
BMI	−0.19	0.43	0.19	0.42	0.03	0.89	0.17	0.47	0.3	0.21	0.52	0.02 *
FTA	0.26	0.28	0.27	0.25	0.48	0.03 *	0.57	0.01 **	0.48	0.03 *	0.55	0.01 *
Knee extension strength	−0.19	0.41	−0.26	0.26	0.34	0.14	0.01	0.96	−0.19	0.42	−0.26	0.26

**: *p* < 0.01, *: *p* < 0.05.

**Table 5 sensors-24-05137-t005:** Partial correlation with adjustment for the body mass index and femorotibial angle between fear-avoidance beliefs and co-contraction during each movement.

Variable	CCR											
	Gait				Stair Ascent				Stair Descent			
	Medial Muscles		Lateral Muscles		Medial Muscles		Lateral Muscles		Medial Muscles		Lateral Muscles	
	r	*p*-Value	r	*p*-Value	r	*p*-Value	r	*p*-Value	r	*p*-Value	r	*p*-Value
TSK-11 score	0.11	0.68	−0.02	0.95	0.2	0.44	0.47	0.05	0.54	0.02 *	0.38	0.12
PCS score	−0.13	0.6	0.03	0.89	0.27	0.29	0.26	0.29	0.31	0.21	0.24	0.34

*: *p* < 0.05.

## Data Availability

Data are contained within the article.

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
