# Peer review of "Relationship between Fear-Avoidance Beliefs and Muscle Co-Contraction in People with Knee Osteoarthritis"

_sensors, 2024, doi:10.3390/s24165137_

Round 1

Reviewer 1 Report

Comments and Suggestions for Authors

Comments to the Author

The manuscript reports on the relationship between fear-avoidance beliefs and muscle co-contraction in individuals with knee OA, and finds that kinesiophobia is associated with excessive muscle co-contraction during stair descent. The manuscript presents interesting results; however, there are some concerns outlined below, based on which I recommend revisions to the manuscript.

Material and Methods:

Study Design and Participants: L87-; 

• Please specify the total number of subjects who participated in the present study in the text in the Material and Methods section.

• This study includes only participants with knee OA. Is the reason for excluding normal controls specified in the exclusion criteria?

• If possible, please include information on the thickness of the subcutaneous fat in the thigh in the results section. The thickness of subcutaneous fat is a critical factor affecting the surface EMG signal.

Data Collection and Processing: L122-; 

• Please provide more details about the electrodes used to record the surface EMG, such as their size, shape, and inter-electrode distance. This information would be beneficial to the reader.

• In this study, the surface EMG signals were filtered between 20-500 Hz to remove noise and artifacts. However, the surface EMG, consisting of motor unit action potentials, can also contain components at frequencies below 20 Hz. Please provide the rationale for prioritizing the removal of noise and artifacts in this study.

• What is the estimated crosstalk between the quadriceps muscles (i.e., VL-VM) and the hamstring muscles (i.e., BF-ST) in this study?

• In Figure 1-a, is the muscle labeled "Semimembranosus" actually "Semitendinosus"?

Results:

• Were there any sex differences in the results of this study, particularly regarding muscle co-activation and fear-avoidance beliefs? Including information on sex differences in the results or discussion sections would be beneficial to the reader.

Discussion:

• Greater focus is needed on the possible mechanisms underlying the association between kinesiophobia and excessive co-contraction in participants with knee OA during stair descent. In this study, did you conduct balance testing or assess fear of falling using the Fall Efficacy Scale (FES) or the Activity-Specific Balance Confidence (ABC) scale? Additionally, was leg muscle strength test performed? An in-depth discussion of the neural mechanisms leading to excessive co-contraction in knee OA patients is critical for developing effective rehabilitation programs aimed at achieving functional recovery.

Limitations: L347-

• The authors calculated the required sample size (n>17) for this study, and the number of subjects included exceeds this estimate. Therefore, statistically, it does not seem necessary to describe the sample size as a limitation.

Reviewer 2 Report

Comments and Suggestions for Authors

This study investigated the relationship between fear-avoidance beliefs and muscle co-contraction in patients with knee osteoarthritis. Knee osteoarthritis is a common disease that occurs with aging, and this is a good paper to show readers the relationship between muscle formation and n fear-avoidance beliefs.

Please consider the following:

1. lines 45-46. Please provide citations for all presented sentences.

2. Wouldn’t it have been possible to check the pain index of the subjects according to the inclusion criteria and classify them? There appears to be a significant difference in walking ability depending on the pain.

3. 2.2. Please provide citations for the testing method of the Testing Protocol.

4. Please indicate the reliability of the evaluation equipment.

5. It seems that abbreviations need to be organized. It appears repeatedly.

6. Please provide details on how to present the effect size for 17 participants. Prior research or pilot test, applied variables, etc.

7. Please add to the review the benefits that muscle co-contraction can provide to patients with knee osteoarthritis.

Round 2

Reviewer 1 Report

Comments and Suggestions for Authors

Revised manuscript was considerably improved, and I have no additional comments.